# Revisiting Mean Flow and Mixing Properties of Negatively Round Buoyant Jets Using the Escaping Mass Approach (EMA)

**Aristeidis A. Bloutsos** [1,2] **and Panayotis C. Yannopoulos** [1,*]

[1]   Environmental Engineering Laboratory, Department of Civil Engineering, University of Patras,
      265 04 Patras, Greece; abloutsos@uniwa.gr
[2]   Hydraulics & Geotechnical Engineering Division, Department of Civil Engineering,
      University of West Attica, 122 43, Athens, Greece
*    Correspondence: yannopp@upatras.gr; Tel.: +30-2610-996527

**Abstract:** The flow formed by the discharge of inclined turbulent negatively round buoyant jets is common in environmental flow phenomena, especially in the case of brine disposal. The prediction of the mean flow and mixing properties of such flows is based on integral models, experimental results and, recently, on numerical modeling. This paper presents the results of mean flow and mixing characteristics using the escaping mass approach (EMA), a Gaussian model that simulates the escaping masses from the main buoyant jet flow. The EMA model was applied for dense discharge at a quiescent ambient of uniform density for initial discharge inclinations from 15° to 75°, with respect to the horizontal plane. The variations of the dimensionless terminal centerline and the external edge's height, the horizontal location of the centerline terminal height, the horizontal location of centerline and the external edge's return point as a function of initial inclination angle are estimated via the EMA model, and compared to available experimental data and other integral or numerical models. Additionally, the same procedure was followed for axial dilutions at the centerline terminal height and return point. The performance of EMA is acceptable for research purposes, and the simplicity and speed of calculations makes it competitive for design and environmental assessment studies.

**Keywords:** negatively buoyant jets; inclined turbulent jets; brine; curvilinear coordinate system; turbulent mixing; integral model

## 1. Introduction

The effluents of a wastewater treatment plant or power plant cooling waters are usually discharged in large water bodies like ocean, lakes, etc. Similarly, the gases from cooling towers and chimneys are emitted in the atmosphere. In the case of the latter, the effluents' density is less than the ambient density. Consequently, the flow has positive buoyancy, forming turbulent buoyant plumes that affect the receiver. These flows have been extensively investigated through the years [1–9]. In contrast, when the discharge fluid is denser than the ambient, the buoyant plumes initially have negative buoyancy, so they are called negatively buoyant jets. In the atmosphere, the emission of dense gases like chlorine, hydrogen fluoride or liquefied natural gas (LNG) occurs when they accidentally leak to the air environment (bounded or unbounded) [10,11]. These gases are usually toxic and/or flammable, causing serious adverse effects to human health, nonhuman biota and generally disasters. The accurate and fast prediction of the flow via mathematical models is crucial in evacuating affected areas and rescuing victims [12,13] and extremely important for the risk assessment of industries and storage areas that handle such hazardous materials [14–16].

Other types, and probably the most common of the negatively buoyant flows, are the high-salinity concentrated effluents (brines) from desalination plants when these are discharged into the sea. These effluents are usually issued through positively inclined orifices, rise upward to a maximum height and then fall downward to sea bottom, due to negative buoyancy, forming gravity currents. During the desalination process, seawater intakes into the plant and its salinity are reduced. As a result, potable water is produced to supply coastal populations, while high saline brines need to be disposed back into the sea as byproduct of the process [17]. Salinity measurements at brines area were usually 1–10% above ambient levels [18]. In addition, brine discharges may contain chemical substances (biocides, antiscalants, coagulants, etc.) that are used during the desalination process [18–20]. This situation affects the ambient water body. Several campaigns are reported in the literature [19,21–25], studying the environmental impacts of high-salinity effluents. High salinity discharges affect benthic communities as spreading over the seabed [18,20,24,26–28].

Rising water demands of the last few decades have increased the number of desalination plants, worldwide and obviously brine production [17,29]. Beyond the alternative technologies of a reduction of brine production [30,31], the safe disposal via outfalls remains the most reliable part of a desalination system, in order to achieve rapid dilution of the effluent [18,20,26].

As a result, the engineering importance of designing well performance diffusers of brine discharges has attracted the interest of many researchers. A lot of researchers examined the vertical discharge of negatively buoyant jets. The earliest experimental studies were made by Turner [32] and Abraham [33]. Bloomfield and Kerr [34] proposed a theoretical model of axisymmetric turbulent fountains testing its predictions conducting experimental measurements, and Kaminski et al. [35] experimentally studied the entrainment of negatively buoyant jets, reporting that entrainment is reduced due to negative buoyancy. Recently, Baddour and Zhang [36] and Ahmad and Baddour [37] focused on the validity of Boussinesq's approximation on round turbulent hypersaline fountains. From early experimental work, it became apparent that inclined negatively buoyant jets achieve better mixing characteristics [38–40]. Consequently, beyond the cases of vertical brines, researchers focused on inclined negatively buoyant jets. The earliest experimental research study on inclined negatively buoyant jets conducted by Zeitoun et al. [38] on initial discharges angles of 30°, 45°, 60° and 90° to the horizontal in stagnant ambient, reporting that, when the initial inclination was 60°, better mixing characteristics were obtained, and the trajectory of the effluent was maximized comparing to vertical discharges. The case of 60° to the horizontal in stagnant ambient was studied in more detail by Roberts and Toms [40] and Roberts et al. [41]. Meanwhile, Lane-Serff et al. [42], using a shadowgraph technique, conducted experiments on dense jets issuing initially from 45° and 60°. The same technique has been used by Lindberg [43], who conducted experiments on dense jets with initial inclination angles 30°, 45° and 60°. Bloomfield and Kerr [44] experimentally investigated the behavior of inclined turbulent fountains in a homogenous and calm environment. They determined the maximum height of the external boundary of fountains of initial angles ranged from 30° to 90° to the horizontal. They observed that the maximum height was increasing with initial angle increase, up to approximately 80°, and then decreasing. Several geometric characteristics of inclined negatively buoyant jets are reported by Cipollina et al. [45] for 30°, 45° and 60° and Ferrari and Querzoli [46,47] for initial inclinations ranged from 45° to 90°. Beyond geometrical characteristics, Nemlioglu and Roberts [48] also obtained axial dilution at the impact point for initial angles ranged from 15° to 90°. Kikkert et al. [49] predicted the behavior of negatively buoyant discharges in stagnant ambient via analytical solutions from an integral model and, using optical experimental techniques (light attenuation (LA) and laser-induced fluorescence (LIF)), compared their findings for initial discharge angles 0° to 75°. Shao and Law [50], Lai and Lee [51] and Jiang et al. [52] combined particle image velocimetry (PIV) and laser induced fluorescence (LIF) techniques to study inclined negatively buoyant jets. Shao and Law [50] captured the velocity and concentration fields of 30° and 45° inclined dense jets, and then, they obtained characteristic geometrical features for every case. Lai and Lee [51] investigated dense jets of various initial angles in a stationary ambient and Jiang et al. [52] studied the discharge of 30° and 45°

inclined dense jets in shallow coastal waters, reporting the existence of three different mixing regimes. Papakonstantis et al. [53,54] experimentally studied inclined turbulent round jets with negative buoyancy discharging in a calm homogeneous fluid, providing geometrical and mixing characteristics for discharge angles from 45° to 90° to the horizontal. The same experimental apparatus was used by Nikiforakis et al. [55] to study negatively buoyant jets discharging at 30°, 45° and 60° and, more recently, by Papakonstantis and Tsatsara [56,57]. The effect of nozzle inclination was studied experimentally by Oliver et al. [58,59], Abessi and Roberts [60] and Crowe et al. [61,62]. Oliver et al. [58,59] used the LIF technique, whil Crowe et al. [61,62] used the particle tracking velocimetry (PTV) flow visualization technique to study experimentally dense jets issuing from initial angle ranged from 15° to 75° in a homogenous calm ambient. Abessi and Roberts [60] used a three-dimensional LIF (3DLIF) to appraise the major flow characteristics from 15° to 85° dense jets in a homogenous still environment.

Integral models are used for the theoretical investigation of inclined buoyant flows. The integral models are based on either on Eulerian or Lagrangian approach [2,8,63]. The closure of integral models is based on either the entrainment concept [64] or the jet spreading rate assumption of the mean axial velocity and concentration fields [9,33]. A major disadvantage of the entrainment assumption is that entrainment does not have a global behavior for positive and negative buoyant jets. To overcome the disadvantage, Kaminski et al. [35] proposed a universal entrainment relationship, while Papanicolaou et al. [65] a reduced entrainment coefficient.

The two most widely used integral models are CorJet [7] and the Lagrangian model JETLAG/VISJET [8,66]. These models were initially developed to predict discharges of positive buoyancy. Jirka [67] applied CorJet to investigate inclined negatively buoyant discharges into stagnant homogenous ambient, and Lai and Lee [51] used JETLAG/VISJET incorporating a reduced entrainment coefficient for dense jets.

The simulation of negatively buoyant discharges appears to be more complicated than that of positively buoyant discharges. Lane-Serff et al. [42], Kikkert et al. [49], Ferrari and Querzoli [47] reported the phenomenon of fluid portions that detached from the main flow of the buoyant jet moving vertically upwards causing gravitational instabilities and consequently destroying the axial symmetry of transverse concentration and velocity profiles [49–52,54,60,68–73]. This situation is favored by weak or sometimes negative values of transverse velocity in the concave region of negatively buoyant jets, as reported experimentally by Crowe [69] and theoretically by Yannopoulos and Bloutsos [74].

Although integral models are generally accepted as reliable tools to simulate inclined buoyant jets, the last years there is a notable trend of using computational fluid dynamics (CFD) to study the behavior of these flows. Vafeiadou et al. [75] studied inclined negatively buoyant jets using the commercial software ANSYS CFX and the shear stress transport (SST) model for turbulence closure. Comparing their findings to available experimental data, they concluded a slight underestimation of terminal rise height and a considerable underestimation of the return point. Additionally, Oliver et al. [76] used ANSYS CFX to solve Reynolds averaged Navier-Stokes (RANS) equations to investigate the geometrical and mixing characteristics of inclined negatively buoyant jets employing for turbulence closure the standard form of $k$-$\varepsilon$ model and a calibrated $k$-$\varepsilon$ model through adjustment of the turbulent Schmidt number in the tracer transport equation. Gildeh et al. [77] used the open-source CFD code called OpenFOAM to evaluate the accuracy of seven turbulence models in simulating wall thermal or saline buoyant jets, concluding that realizable $k$-$\varepsilon$ and Launder–Reece-Rodi (LRR) closure turbulence models were found to be the most accurate and capable of accurately modeling thermal and saline wall jets discharged into stationary ambient water. Gildeh et al. [78,79]) studied the behavior of 30° and 45° inclined dense jets in stationary ambient using OpenFOAM. They applied several closure-turbulence-models to evaluate the accuracy of CFD predictions of geometrical and mixing characteristics. They reported that differences occur among the models with realizable $k$-$\varepsilon$ and LRR tending to be more accurate among them. The realizable $k$-$\varepsilon$ was used by Ardalan and Vafaei [80] too, to numerically model thermal-saline effluent discharging at an angle of 45° in a homogenous and still environment. More recently, large eddy simulations (LES) were used by Zhang et al. [81,82], Cintolesi et al. [83] and Jiang et al. [72] to

study in detail the geometrical mixing and turbulent characteristics of inclined dense jets, reporting that there are still challenges to simulate accurately the mixing behavior of inclined dense jets.

Briefly, the flow field of an inclined buoyant jet discharged into a stationary ambient fluid of uniform density is described by the initial flux of volume $\mu_0 = A_0 w_0$, the initial flux of momentum $m_0 = A_0 w_0^2$ and the initial flux of buoyancy $\beta_0 = g_0' A_0 w_0 c_0$, where $w_0$ is the initial discharge velocity, $A_0 = \pi d_0^2/4$ is the area of the exit nozzle, $d_0$ is the diameter of the exit nozzle, $c_0$ is the initial concentration of tracer, $g'_0 = g(\rho_a - \rho_0)/\rho_0$ is the apparent gravitational acceleration, $g$ is the gravitational acceleration, $\rho_0$ and $\rho_a$ are the densities of the fluid at the jet exit and the ambient, respectively. Yannopoulos and Bloutsos [74], following Shao and Law [50], stated that negative buoyancy occurs when the vertical component of the initial momentum of the buoyant jet is zero, or the jet discharges with purely horizontal momentum and $\rho_0 > \rho_a$ so $g'_0 < 0$ or alternatively when the vertical component of the initial momentum is non-zero, and the resulting buoyancy force acts in the opposite direction as the initial motion of the buoyant jet. Additionally, dimensional analysis shows that the behavior of such a flow field is described by three independent non-dimensional variables: the initial densimetric Froude number $F_0 = w_0 \sqrt{|g'_0| d_0}$, the initial inclination angle $\theta_0$ measured with respect to the horizontal plane and the dimensionless axial distance $\Xi = \xi/l_m = (\xi/d_0) F_0^{-1}$, where $\xi$ is the axial geometrical distance and $l_m$ is the characteristic length scale $l_m = d_0 F_0$ that is usually employed to make geometrical distances non-dimensional [74]. All other parameters are either constants or variables depending upon independent ones. The constant parameters have the same values both for vertical or inclined buoyant jets of either positive or negative buoyancy. The only extra parameter included in inclined buoyant jets with negative buoyancy is the parameter $\Lambda$, which is defined in Chapter 2. More details about the description of the flow field of an inclined buoyant jet discharged into a stationary ambient fluid of uniform density can be found at the relevant papers of Yannopoulos [9] and Yannopoulos and Bloutsos [74]. As the present paper revisits the previously published work regarding the negatively round buoyant jet using the EMA model [74], the authors effort is to incorporate in the current paper an overall review of all the experimental data and numerical results of such problems.

The present paper determines the geometric and mixing characteristics of an inclined round negatively buoyant jet using the innovative mathematical model, called escaping mass approach (EMA) [74], which incorporates the exact description of the flow field through a curvilinear coordinate system, the application of a Gaussian integral model including second order terms [9]) and the consideration of escaping masses from the concave region of the inclined buoyant jet [74]. As the curvilinear coordinate system includes the main flow axis $\xi$ and the transverse axis $r$, it enables the exact integration of the transverse Gaussian profiles of the axial velocity and concentration. Therefore, the calculation of the volume, momentum and buoyancy fluxes are computed exactly along the curvilinear path of the main flow of the buoyant jet.

The EMA model was applied for negatively buoyant jets of initial inclinations in the range $15° \leq \theta_0 \leq 75°$ and initial Froude number range $4 < F_0 < 100$ in a stationary and homogenous ambient environment. EMA's performance was checked by comparing geometric (centerline and external edge terminal height, horizontal location of centerline terminal height and return point, centerline length to terminal height and centerline and external edge length to return point) and mixing (minimum dilution at centerline terminal height and return point) characteristics to available experimental data, numerical and integral model predictions. The good agreement of the results of the EMA model with experimental data obtained from the international literature confirms the reliability of the model. In addition, the better performance in comparison to other integral or numerical model results makes the EMA model appropriate in the design of systems for the disposal of liquid, thermal and gas waste under these conditions. It could also be used for the verification of complex numerical models, but also for laboratory studies of related models.

An inclined negatively turbulent buoyant jet of density $\rho_0$ is discharged upwards with initial velocity $w_0$ from a round nozzle of size $d_0$ with an initial inclination $\theta_0$ to the horizontal plane into

a quiescent ambient fluid of uniform density $\rho_a$ ($\rho_0 > \rho_a$). At the first stage, the buoyant jet moves upwards, due to its initial momentum and the mixing with ambient fluid starts. Meanwhile, the negative buoyancy decreases the momentum flux causing its deceleration. The jet rises to a terminal (maximum) height, where the vertical component of momentum becomes zero, and then it falls as a positively buoyant jet to the initial discharge elevation. The general configuration of such a flow is shown in Figure 1, related to a Cartesian coordinate system ($O$, $x$, $y$, $z$). The flow is assumed to be steady state. The flow centerline (trajectory) is defined as the location of maximum velocity or concentration at each cross-section of the flow. In Figure 1, $y$ and $z$ are the horizontal and vertical distances and subscripts $c$, $e$, $t$ and $r$ denote the jet centerline, the external boundary, the terminal rise height and the return point to the source elevation, respectively. Following these definitions $z_{ct}$ and $z_{et}$ are the corresponding terminal heights of the centerline and the external boundary of the jet, which occur at a horizontal distance $y_{ct}$ from the source. The horizontal location of the return point of the jet axis is $y_{cr}$ and the corresponding location for the external boundary is $y_{er}$. Additionally, $S_{ct}$ and $S_{cr}$ are the axial dilutions at terminal rise height and the return point, correspondingly. Previous studies [32,38,40,45,49,50,53,54,56–62,81] have verified that the geometrical quantities non-dimensionalized by $d_0$, and dilutions are proportional to $F_0$ for a specific angle $\theta_0$.

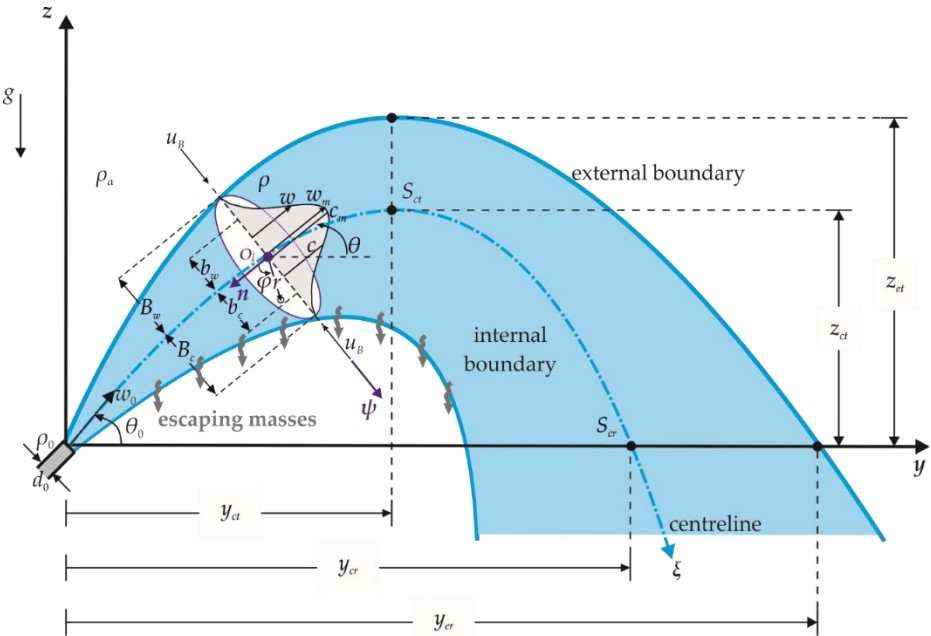

**Figure 1.** Definition sketch of flow of an inclined turbulent negatively buoyant jet.

## 2. Materials and Methods

The escaping mass approach (EMA) model [74] is an integral model that describes the flow and mixing fields of inclined round or plane buoyant jets issuing into stationary ambient environment of uniform density. For the case of round buoyant jets, the conservation partial differential equations (PDE) of continuity, momentum and tracer have been formulated in a curvilinear cylindrical coordinate system, which is relative to the corresponding curvilinear orthogonal Cartesian coordinate system for plane buoyant jets [84]. EMA simulates the reported phenomenon ([42,47,49–52,54,60,68–73] of fluid portions that detach from the main flow of the buoyant jet, especially from the inner boundary, and move vertically upwards (or downwards for brines), causing gravitational instabilities, and consequently destroying the axial symmetry of transverse concentration and velocity profiles. EMA calculates the vertical velocity $w_B$ of the escaping masses from the buoyant jet field, employing the general definition of the local Richardson number for plume flows [9]. The concentration of these masses $c_B$ is estimated as a portion $\Lambda$ of its centerline value $c_m$. The value of $\Lambda$ coefficient was adopted equal to

0.34 [74], where $\Lambda$ includes the value $\pi/4$ for round buoyant jets) for all cases in the inclination range $-75° \leq \theta_0 \leq -15°$ (or $15° \leq \theta_0 \leq 75°$ of negatively buoyant jets). Additionally, EMA is equipped with the second order approach (SOA), to include the variable second-order effect of turbulence to the mean flow properties [9].

EMA's development [74] is based on PDEs of volume momentum and tracer's conservation, concerning the Reynolds approximation for mean flow and mixing properties and their fluctuations in steady-state conditions. Additionally, the usually accepted assumptions of (a) Boussinesq's approximation made for small initial density differences; (b) Prandtl's boundary-layer approximation; (c) negligible molecular viscosity terms and (d) no swirl are adopted. Regarding flow symmetry with respect to the centerline $\xi$ axis of the curvilinear coordinate system $O_l(r, \varphi, \xi)$, the governing equations of the model are:

Continuity

$$\frac{\partial(rw)}{\partial\xi} + \frac{\partial(rhu)}{\partial r} = 0 \tag{1}$$

$\xi$–momentum

$$\frac{\partial}{\partial\xi}\left[r\left(w^2 + w'^2 + \frac{p_d}{\rho_0}\right)\right] + \frac{\partial}{\partial r}\left[rh\left(uw - \frac{\tau_{r\xi}}{\rho_0}\right)\right] + r\sin\varphi\left(uw - \frac{\tau_{r\xi}}{\rho_0}\right)\frac{d\theta}{d\xi} = g'_0 rhc\sin\theta \tag{2}$$

$\psi$–momentum

$$\frac{\partial}{\partial\xi}\left[(r\sin\varphi)\left(uw - \frac{\tau_{r\xi}}{\rho_0}\right)\right] + \frac{\partial}{\partial r}\left[(r\sin\varphi)h\left(u^2 + \frac{p_d}{\rho_0}\right)\right] - r\left(w^2 + w'^2 + \frac{p_d}{\rho_0}\right)\frac{d\theta}{d\xi} = -g'_0 rhc\cos\theta \tag{3}$$

Mass tracer

$$\frac{\partial}{\partial\xi}[r(wc + w'c')] + \frac{\partial}{\partial r}[rh(uc + u'c')] = S_1 \tag{4}$$

where $w$, $u$ are the mean velocity components in the directions $\xi$ and $r$, respectively and $w'$, $u'$ are their corresponding turbulent fluctuations; $r = (n^2 + \psi^2)^{1/2}$ is the transverse (radial) distance; $h = 1 + r\sin\varphi\, d\theta/d\xi$ is the scale factor of the coordinate system, $\theta$ is the local inclination angle of the $\xi$ axis, $c = (\rho_a - \rho)/(\rho_a - \rho_0)$ is the local mean concentration and $c'$ the corresponding turbulent fluctuation; $\rho$ is the local mean fluid density; $w'^2$, $w'c'$, $u'c'$ are the local fluxes due to turbulent fluctuations of $w$, $u$, $c$; $\tau_{r\xi}$ is the mean turbulent shear stress; $p_d$ is the dynamic pressure; $S_1 = \partial(hq_r)/\partial r$ is a source term for the escaping masses from the flow field of the buoyant jet; and $q_r$ is the escaping load of masses per unit area in the $r$ direction.

The flow field is further described when Equations (1)–(4) satisfy the following boundary conditions:
On the jet axis ($r = 0$)

$$u = 0, w = w_m, c = c_m, \tau_{r\xi} = u'c' = q_r = 0, p_d = p_{dm} \tag{5}$$

On the jet boundary ($r = B_w$)

$$u = U_B, w = c = \tau_{r\xi} = u'c' = 0, q_r = q_B\sin\varphi, p_d = p_{de} \tag{6}$$

where $U_B = -u_B - w_B\cos\theta\sin\varphi$. Based on experimental measurements, the region between the nominal half-width of a buoyant jet, $b_{0.5}$, and the total half-width, $B_w$, is characterized by intermittencies of the velocity field (see Yannopoulos & Bloutsos 2012 [74]). Therefore, our assumption is that all this region could contribute to escaping masses. The velocity $w_B$ becomes zero at $r = b_{0.5}$, while has a positive value at the boundary of the buoyant jet ($r = B_w$). The escaping masses from the convex boundary enter the buoyant jet and are thus swept by the main flow. However, only masses from the concave boundary manage to escape when the entrainment of the buoyant jet becomes very weak or negative.

Consequently, in addition to Equation (6), on the jet boundary ($r = B_w$) further boundary conditions are applied:

$$q_B = \begin{cases} 0 & \text{for } 0 \leq \varphi \leq \pi \\ w_B c_B \cos\theta & \text{for } \pi < \varphi \leq 2\pi \end{cases} \tag{7}$$

where $w_B$ and $c_B$ are the vertical velocity and concentration of escaping masses; $u_B$ is the entrainment velocity of the inclined buoyant jet at the actual boundary $B_w = d_0/2 + n_w\, b_w$ and $n_w = 2^{-1/2}$ [7,8]).

The main flow of the buoyant jet is assumed that remains symmetrical to the jet $n\xi$ plane. Therefore, the dimensionless mean axial velocity and concentration profiles are assumed Gaussian in the zone of established flow (ZEF):

$$\frac{\phi}{\phi_m} = \exp\left(-\frac{r^2}{b_\phi^2}\right) \tag{8}$$

where $b_\phi = K_\phi \xi$ is the nominal width of the buoyant jet where the property $\phi$ reduces at the $e^{-1}$ of the maximum value $\phi_m$ and $K_\phi$ is the spreading rate coefficient. Thus, $\phi$ stands for either the mean axial velocity $w$ or the mean concentration $c$.

In the zone of flow establishment (ZFE), the EMA model incorporates the advanced integral model (AIM; [85]). Thus, the actual similarity profiles of velocity and concentration are yielded by composing the flow and mixing fields of a group of infinite vertical buoyant jets issued from closely spaced sources on the basis of flux conservation of momentum, buoyancy and kinetic energy for the mean motion and enhancing the dynamic adaptation of the individual buoyant jet axes to the group centreline.

The entrainment velocity $u$ is calculated using the transverse velocity profile $hu/w_m$ that is produced by integrating the equation of continuity (1) with respect to distance $r$ [9,74]:

$$\frac{hu}{w_m} = -K_w\left\{\frac{1}{2}\left(2 + \frac{\xi}{w_m}\frac{dw_m}{d\xi}\right)\left[1 - \exp\left(-\frac{r^2}{b_w^2}\right)\right]\left(\frac{r}{b_w}\right)^{-1} - \frac{r}{b_w}\exp\left(-\frac{r^2}{b_w^2}\right)\right\} \tag{9}$$

According to [74], the vertical velocity $w_B$ of the escaping masses at the inner boundary is estimated as:

$$w_B^2 = \frac{3pg'_0\lambda_{Bp}\lambda_p}{R_p Y_p}\frac{n_w - \sqrt{\ln 2}}{\cos\theta}c_B b_w \tag{10}$$

where $p = 0.144$, $K_w = 0.11$, $\lambda_{Bp} = 1.15$, $\lambda_p = 1.04$, $R_p = 0.3521$ and $Y_p = 1.001$ [74]. Additionally, the concentration $c_B$ of the escaping fluid by the inner boundary is assumed proportional to the corresponding centerline concentration as:

$$c_B = \Lambda c_m \tag{11}$$

where $\Lambda$ is a constant that is determined approximately to 0.34 [74].

## 3. Variation of Basic Parameters

Integration of the PDE of tracer mass (4), under the assumptions (5)–(8) yields the ordinary differential equation (ODE) of tracer mass $\frac{d}{d\xi}\left(\frac{\beta}{g'_0}\right) = -2B_w\left(1 - \frac{\pi}{4}B_w\frac{d\theta}{d\xi}\right)\Lambda c_m w_B \cos\theta$, which gives the longitudinal variation of the slop of the buoyancy flux [74]. Figure 2 shows the variation of the normalized buoyancy flux with respect to the dimensionless distance $\Xi$. It is apparent that the buoyancy flux reduces along centerline path until $\Xi \cong 10$ and then stabilizes to a constant value. This reduction increases via initial inclination angle $\theta_0$ until $\theta_0 \cong 40°$–45° and then decreases up to 75°. Thus, as seen in Section 4, the distance $y_{cr}$ increases until $\theta_0 \cong 40°$–45° and then decreases up to a minimum value approaching zero for vertical discharges.

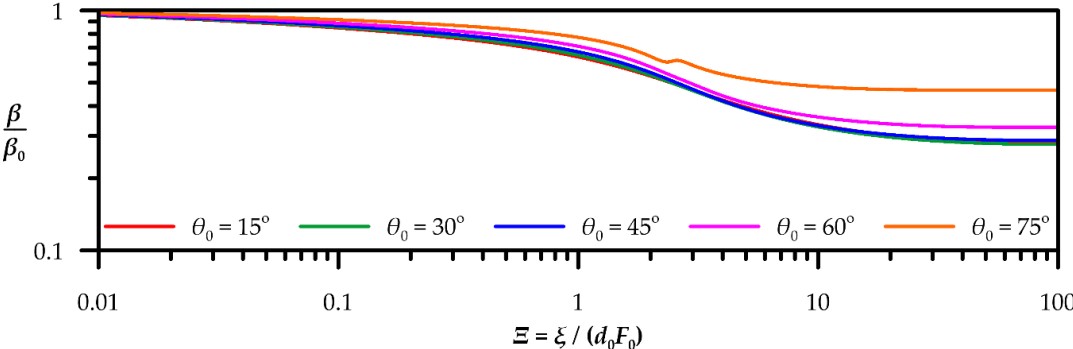

**Figure 2.** Variation of the normalized buoyancy flux $\beta/\beta_0$ with dimensionless distance $\Xi = \xi/(d_0 F_0)$ and initial inclination $\theta_0$.

After the flow bends over, beyond the cross section where the maximum $z_{ct}$ occurs, the buoyancy flux becomes positive. The major part of this region is beyond $\Xi \cong 10$ (Figure 2). Therefore, the stability of the buoyancy flux means that the assumption made regarding the escaping mass has no effect in the region of positive buoyancy flux.

The longitudinal variation of the local angle $\theta$ of the buoyant jet axis to horizontal with respect to the dimensionless distance $\Xi = \xi/(d_0 F_0)$ for various initial inclination angles $\theta_0$ ($\theta_0 = 15°, 30°, 45°, 60°, 75°$) and initial Froude numbers $F_0$ ($F_0 = 5, 20$) is shown in Figure 3. It is apparent that the effect of $F_0$ to $\theta$ is rather insignificant for practical applications.

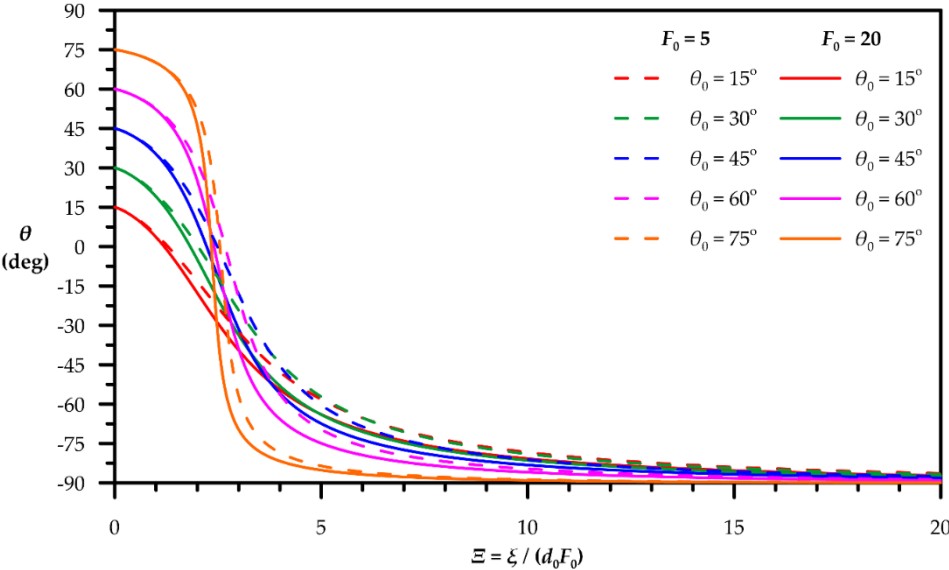

**Figure 3.** Variation of local centerline angle $\theta$ to horizontal with dimensionless distance $\Xi = \xi/(d_0 F_0)$ and initial inclination $\theta_0$ for $F_0 = 5$ and $F_0 = 20$.

## 4. Results and Discussion

PDEs (1)–(4) are integrated with respect to $r$ and $\phi$ on the actual cross-section of a round buoyant jet under the aforementioned boundary conditions, the spreading concept [9,85] and all the appropriate approximations [74]. The integration produces a system of ordinary differential equations (ODE) that is solved using a 4th order Runge-Kutta method [86]. More details about the procedure may be found at the relative paper of Yannopoulos and Bloutsos [74].

In this paper, EMA is applied to predict the mean flow geometrical quantities and mixing characteristics for inclined negatively turbulent buoyant jets for initial inclination angles $15° \leq \theta_0$

$\leq 75°$ and initial densimetric Froude numbers $4 \leq F_0 \leq 100$. For each angle, in the range $15° \leq \theta_0 \leq 75°$, the variation of the non-dimensional geometric quantities of $z_{ct}/d_0$, $z_{et}/d_0$, $y_{ct}/d_0$, $y_{cr}/d_0$ and $y_{er}/d_0$ and mixing characteristics $S_{ct}$ and $S_{cr}$ are calculated for every initial Froude numbers $4 \leq F_0 \leq 100$. For every $\theta_0$, these quantities vary proportionally to the Froude number by a constant $k_i$ (among others: [49,53,54]) as:

$$\frac{z_{ct}}{d_0} = k_1 F_0, \ \frac{z_{et}}{d_0} = k_2 F_0, \ \frac{y_{ct}}{d_0} = k_3 F_0, \ \frac{y_{cr}}{d_0} = k_4 F_0, \ \frac{y_{er}}{d_0} = k_5 F_0, \ S_{ct} = k_6 F_0, \ S_{cr} = k_7 F_0 \qquad (12)$$

where $k_i$ ($i = 1, \ldots, 7$) is the proportionallity constant. The results are compared to experimental data available in the literature published by several researchers, as well as to other numerical models predictions such as CorJet, VisJet, the reduced buoyancy flux (RBF) model by Oliver et al. [58], modified RBF by Crowe et al. [61] and analytical solutions proposed by Kikkert et al. [49]. Available data from CFD analysis [76,78,81,82] are also used for comparisons.

　　Figure 4 shows the variation of dimensionless centerline terminal rise height $z_{ct}$ to the initial inclination angle. EMA's prediction is compared to available experimental data. The terminal rise height EMA passes through the scatter of experimental data following their trend. The latter is more obvious if a polynomial fit to experimental data [45,47,49–51,54,57–59,61,73,81,87,88] is made. For this purpose, a third degree polynomial is fitted to experimental data using the least square method. The equation of the polynomial is presented in Table 1. In the range $15° \leq \theta_0 \leq 75°$ EMA's prediction is very close to the polynomial fit. It must be noted that EMA predicts the centerline as the midpoint of the round cross-section. The same procedure is followed by Bosanquet et al. [87] and Bashitialshaaer et al. [88]. Instead, Cipollina et al. [45], Kikkert et al. [49], Ferrari and Querzoli [47], Shao and Law [50], Lai and Lee [51], Oliver et al. [58], Nikiforakis et al. [89], Crowe et al. [61], Ramakanth [73], and Zhang et al. [81] determine the position of centerline as the point where the transverse profile of velocity or concentration is maximized. In particular, this point is shifted towards the external edge, moving away from the midpoint of cross-section. As the initial inclination angle is rising, the deviation of actual transverse profile is deviates intensively from the Gaussian presenting this difference, while, when the initial inclination reaches the vertical, this deviation is reduced, and EMA's prediction coincides with the experimental data. Thus, the deviation between EMA's prediction and the experimental data in the range $20° < \theta_0 < 70°$ is reasonably expected. In Figure 4b, EMA's prediction of centerline terminal rise height is compared to other models results. For $\theta_0 < 30°$ all models almost coincide. For $\theta_0 > 30°$, Kikkert's analytical solution and the RBF model are diverging upwards calculating higher values than EMA's and modified RBF's predictions that approximately coincide up to 75°. The commercial packages CorJet and VisJet are underestimating the centerline terminal rise height for $\theta_0 > 30°$. In Figure 4b, the results of CFD analysis from Gildeh et al. [78,79] Oliver et al. [76] and Zhang et al. [81,82] for a variety of simulations are included. CFD's results are for the most usual initial inclination angles of 15°, 30°, 45° and 60°. For 15°, all CFD's results coincide, showing that the option of RANS or the most sophisticated LES approach does not have major effect on the results. There are significant differences among CFD simulations for $\theta_0 = 30°$, 45° and 60°, where the scatter of the results is maximized for $\theta_0 = 45°$. In any case, EMA predictions are within experimental data.

**Table 1.** Equations of polynomial fits (3rd degree) to experimental data.

| Quantity | Figure | Polynomial Fit |
|---|---|---|
| Centerline terminal height | 2 | $z_{ct}/(d_0F_0) = 1.6275 \times 10^{-1} - 1.2691 \times 10^{-2}\theta_0 + 1.2672 \times 10^{-3}\theta_0^2 - 1.0583 \times 10^{-5}\theta_0^3$ |
| External edge terminal height | 3 | $z_{et}/(d_0F_0) = 4.4066 \times 10^{-1} - 2.1578 \times 10^{-3}\theta_0 + 1.0565 \times 10^{-3}\theta_0^2 - 9.1929 \times 10^{-6}\theta_0^3$ |
| Horizontal distance to terminal height | 4 | $y_{ct}/(d_0F_0) = 6.0332 \times 10^{-1} + 6.7018 \times 10^{-2}\theta_0 - 8.4160 \times 10^{-4}\theta_0^2 + 4.4949 \times 10^{-7}\theta_0^3$ |
| Horizontal distance to centerline return point | 5 | $y_{cr}/(d_0F_0) = 1.3833 \times 10^{0} + 1.0256 \times 10^{-1}\theta_0 - 1.3447 \times 10^{-3}\theta_0^2 + 7.3392 \times 10^{-7}\theta_0^3$ |
| Horizontal distance to external edge's return point | 6 | $y_{er}/(d_0F_0) = 3.6327 \times 10^{0} - 1.0627 \times 10^{-2}\theta_0 + 9.8744 \times 10^{-4}\theta_0^2 - 1.3289 \times 10^{-5}\theta_0^3$ |
| Minimum dilution at terminal height | 7 | $S_{ct}/F_0 = -3.5982 \times 10^{-3} + 2.3515 \times 10^{-2}\theta_0 - 3.9088 \times 10^{-4}\theta_0^2 + 2.1287 \times 10^{-6}\theta_0^3$ |
| Minimum dilution at return point | 8 | $S_{cr}/F_0 = 4.9351 \times 10^{-1} + 1.2654 \times 10^{-2}\theta_0 + 5.0248 \times 10^{-4}\theta_0^2 - 6.5504 \times 10^{-6}\theta_0^3$ |

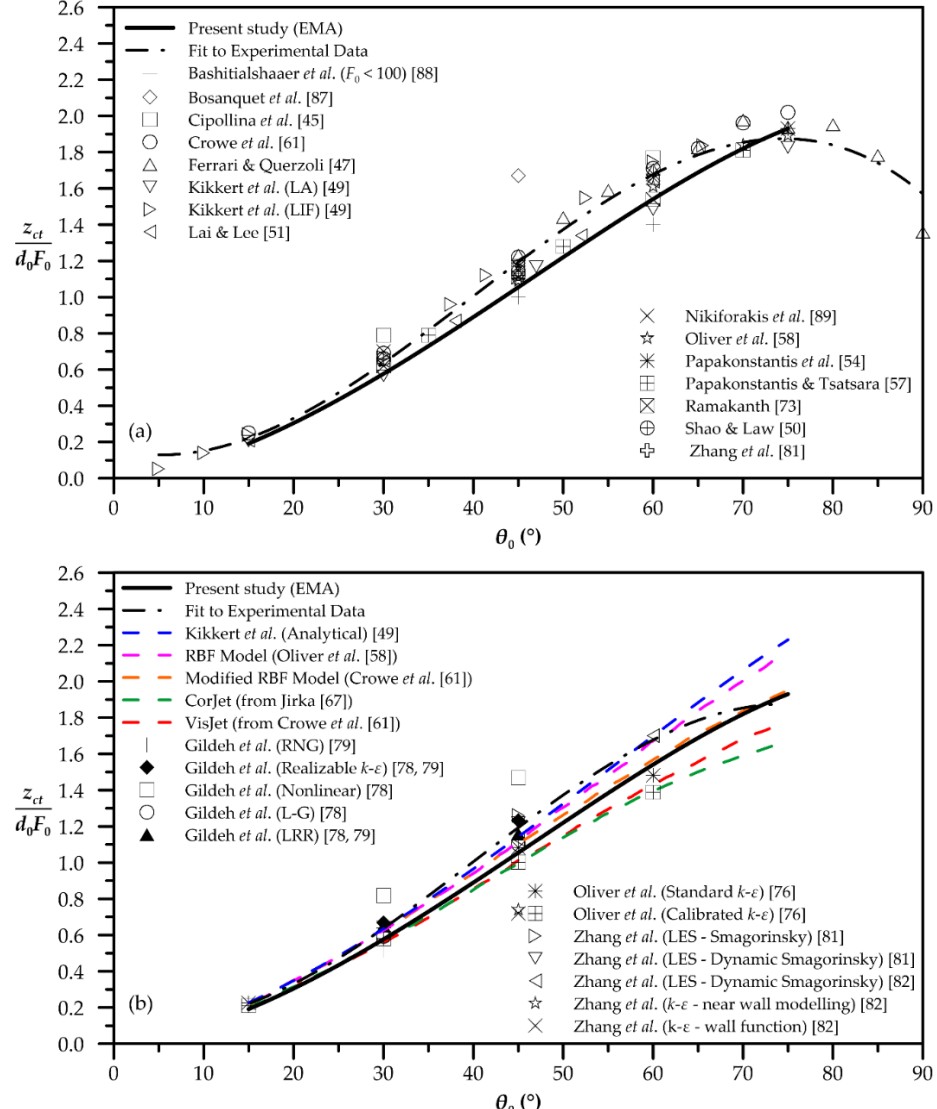

**Figure 4.** Escaping mass approach's (EMA')s prediction of dimensionless centerline terminal height and comparison with (**a**) experimental data, and (**b**) other models' predictions.

The geometric parameter of the final rising height of the external edge (boundary) $z_{et}$ of the dense jets is of great environmental importance, because it indicates whether the mixing processes are taking place under the water surface or not. The dimensionless values of $z_{et}$ that are calculated by the EMA model through the initial inclination angles are shown on Figure 5a,b. In Figure 5a, the EMA's results are compared to previous published experimental data, and in Figure 5b, the performance of EMA is compared to other models. Again EMA, without intense deviation, follows the trend of a third degree polynomial fitted to experimental data [38,40–45,48–51,53,56,58,60,69,73,81,88–93] (Table 1). In Figure 5a, the experimental data of Nemlioglou and Roberts [48] and Bloomfield and Kerr [44] are the upper and lower data of the scatter, respectively. The wide scatter of experimental results is due to the different definitions of the external edge of the buoyant jet among the investigators. Indicatively, Kikert et al. [49] and Oliver et al. [58,78] define the external boundary at a distance twice the concentration spread, while Papakonstantis et al. [54] define this length as 1.5 times the concentration spread. Lai and Lee [51] and Abessi and Roberts [60] define the boundary at a locus where concentration values $c$ are 25% or 10% the maximum concentration ($c_{max}$) at centerline, respectively. Similarly, Jiang et al. [52] and Zhang et al. [81] define the external boundary at $c/c_{max} = 5\%$ and Shao and Law [50] and Gildeh et al. [78] define the external boundary at $c/c_{max} = 3\%$. According to EMA, the outer boundary of the jet is defined at a distance $B_c = d_0/2 + n_c \lambda b_w$, where $b_w$ is the nominal spread width of the velocity field of the buoyant jet, $\lambda$ is the concentration-to-velocity spreading rate coefficient ratio and $n_c$ is the non-dimensional total spread of concentrations of a buoyant that according to profile measurements by Ramaprian and Chandrasekhara [94] and Shao and Law [50] is approximately 1.9. More details can be found at the relative paper of Yannopoulos and Bloutsos [74]. Among EMA's prediction and Kikkerts's solution, the RBF and modified RBF models there are slight differences up to approximately 70°. A little deflection occurs between EMA's and CorJet's prediction for the cutoff level of $c/c_{max} = 3\%$. As VisJet's boundary is defined at 0.25 $c_{max}$, its results are like CorJet's cutoff level of 25%, and obviously differ to the EMA relative values.

In Figure 6a,b, the dimensionless horizontal distance to centerline terminal rise height $y_{ct}$ predicted by EMA is shown. The distance $y_{ct}$ is increasing gradually up to initial angle of 45°, approximately, and then decreases smoothly up to 75°. Similar behavior is observed at experimental data of various previous works demonstrating the good performance of EMA. Excluding the experiments of Bosanquet et al. [87] and Lindberg [43], the experimental scatter is quite narrow, and EMA predictions are within the experimental data for the range $15° \le \theta_0 \le 75°$. CFD analyses provide data only for initial angles of 15°, 30°, 45° and 60°. The scatter of CFD data seems to be wider than experiments. Both CorJet and VisJet diverge from experimental data, underestimating the horizontal distance of centerline peak. EMA's predictions underestimate the corresponding experimental data, following the trend of a third degree polynomial fitted to experimental data [43,45,49–51,53,56,58,61,73,81,87,89] (Table 1).

The prediction of the dimensionless horizontal location of the return point of jet axis to initial level, $y_{cr}$, is shown in Figure 7. EMA's results are compared to several experimental data for various initial inclinations in Figure 7a. Comparing EMA's prediction to experimental fit line [41,45,47–52,54,57,58, 60,61,73,81,89,91,92] (Table 1), it is obvious that EMA follows the fit line, underestimating the return point location in the whole range of $15° \le \theta_0 \le 75°$. The difference is maximized for $\theta_0 = 15°$, decreases approaching approximately $\theta_0 = 60°$ and slightly increases at $\theta_0 = 75°$. Comparing to other models and CFD predictions (Figure 7b), the EMA's predictions seem to be better and rather satisfactory. Additionally, the corresponding variation of the external edge to initial inclination angle is presented in Figure 8. EMA's predictions are compared to the experimental data of Bashitialshaaer et al. [88], Crowe [69], Nikiforakis et al. [89], Oliver [68], Papakonstantis et al. [53], Papakonstatis and Tsatsara [56], Ramakanth [73] and Zeitoun et al. [38], and the modified RBF Model [61]. Again, a third-degree polynomial is fitted to experimental data [38,53,56,68,69,73,88,89] (Table 1). The two models predict quite similar values. The divergence of the EMA model to the experimental fit line is less than 10% for initial angles in the range $15° \le \theta_0 \le 75°$.

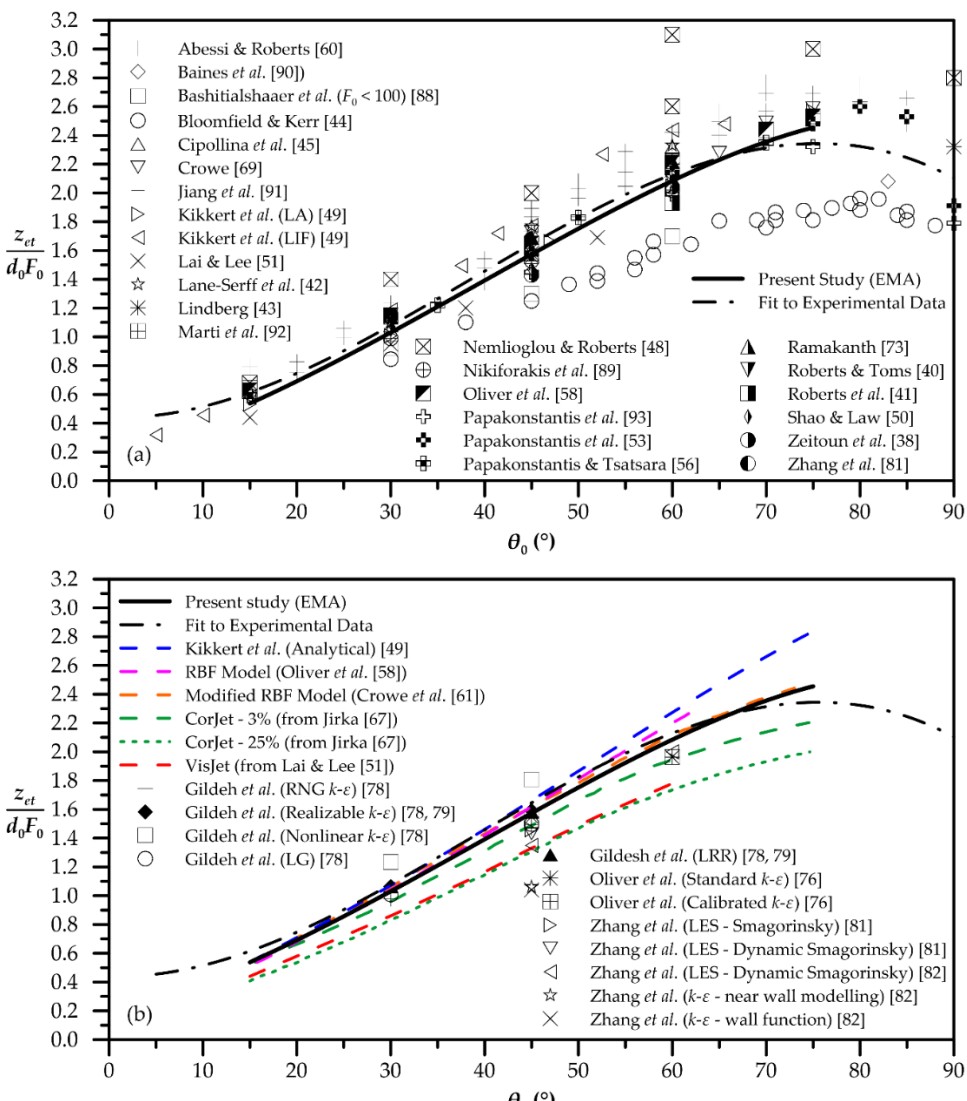

**Figure 5.** EMA's prediction of dimensionless terminal height of external edge and comparison with (**a**) experimental data, and (**b**) other models' predictions.

Figure 9 presents the prediction of the EMA model for axial dilutions at the centerline terminal height ($S_{ct}$). The prediction is compared to available experimental data. EMA's dilution values are overestimated, compared to a third degree polynomial fit [38,40,42,50,51,54,57,58,73,81,91] (Table 1) to experimental data, in the range of initial inclination angles $15° \le \theta_0 \le 65°$. In the range $65° < \theta_0 \le 75°$, the maximum underestimation is 24.0%, which occurred for $\theta_0 = 75°$.

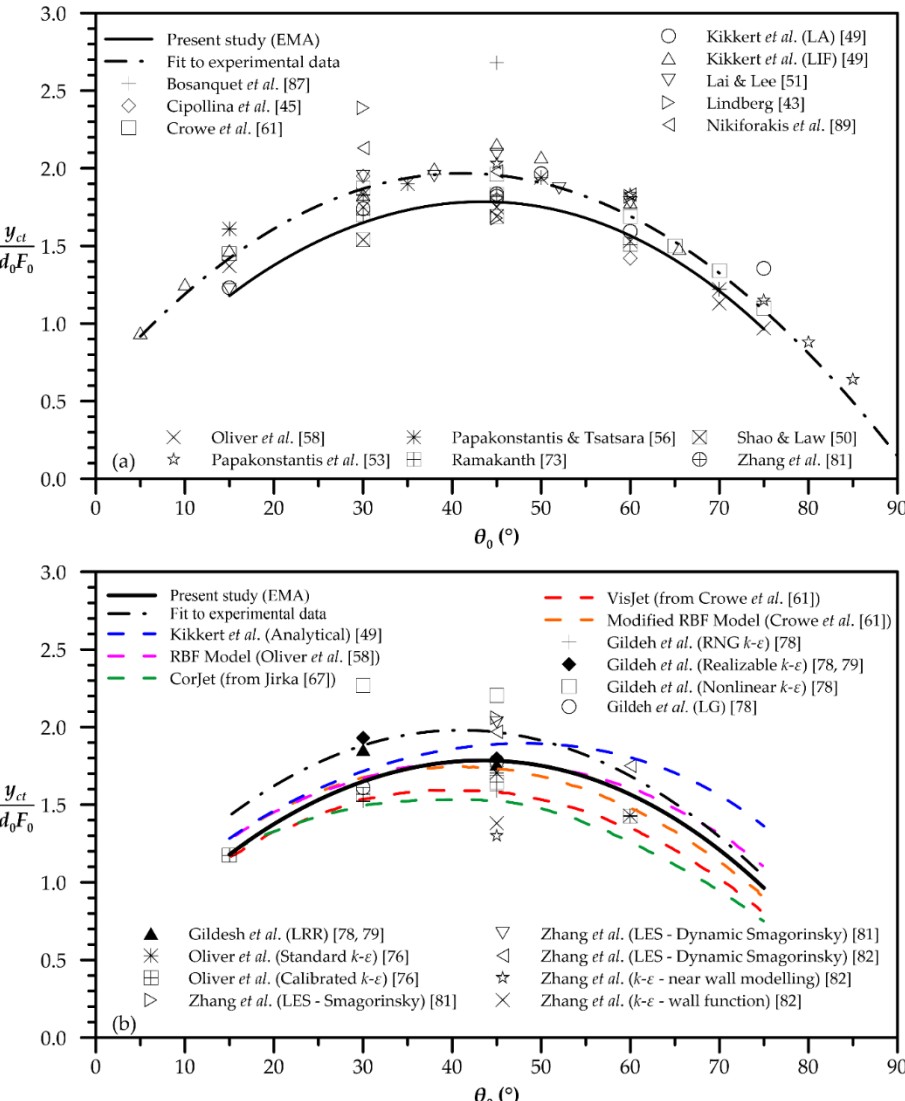

**Figure 6.** EMA's prediction of dimensionless horizontal location of centerline terminal height and comparison with (**a**) experimental data, and (**b**) other models' predictions.

The variation of EMA's axial dilutions at the return point ($S_{cr}$) for initial inclination angles in the range $15° \leq \theta_0 \leq 75°$ is presented in Figure 10. Figure 10a shows the predicted EMA's dilutions compared to available experimental data. A third degree polynomial is fitted to experimental data [40,41,48, 50–52,54,57,59,60,73,81,91,92] (Table 1). EMA's predictions perform a continuous increment, slightly underestimating dilution values in the range of $15° \leq \theta_0 \leq 65°$, while the maximum overestimation is 12.6% for initial inclination angles greater than 65°. Comparing to other model predictions (Figure 10b), it is obvious that EMA has the better overall performance. CFD analysis by Gildesh et al. [78,79] for θ0 = 30° gave similar values approximating experimental data.

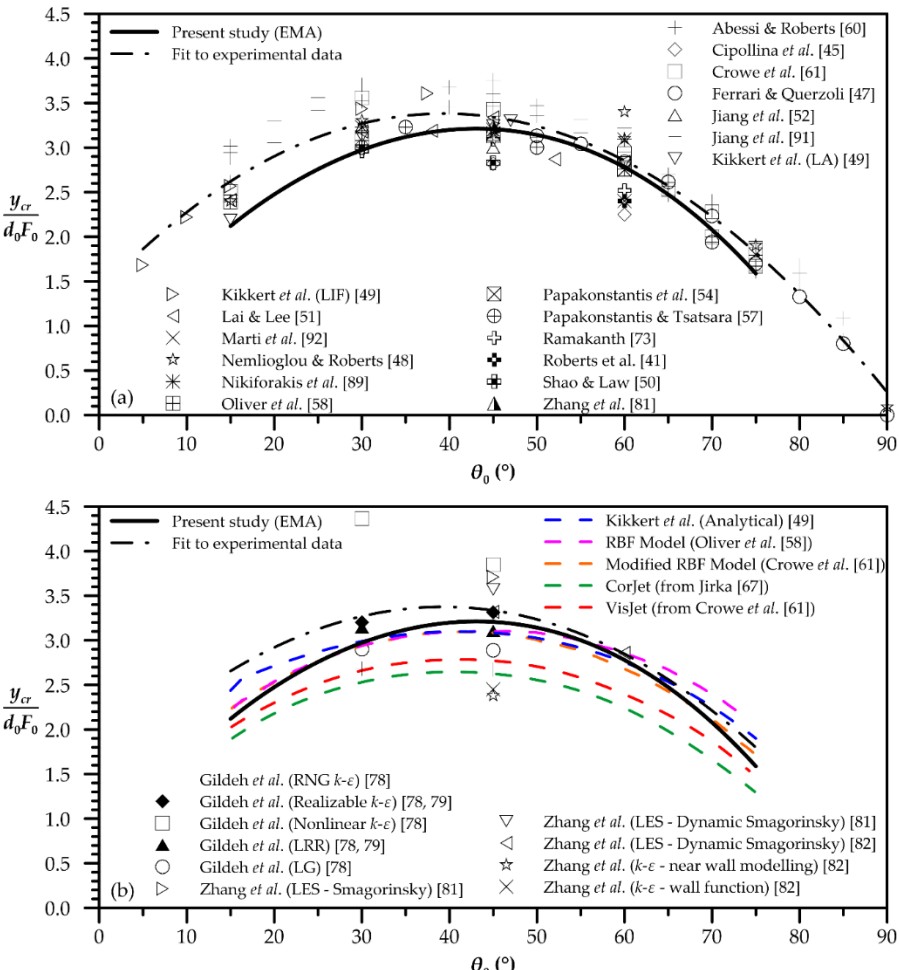

**Figure 7.** EMA's prediction of dimensionless horizontal location of centerline terminal height and comparison with (**a**) experimental data, and (**b**) other models' predictions.

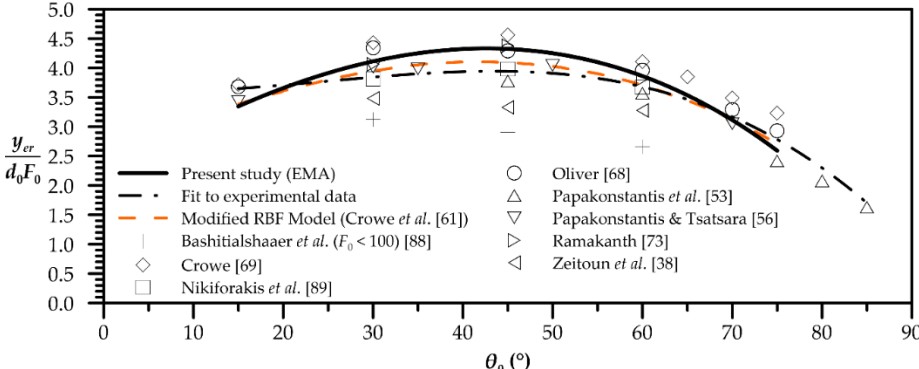

**Figure 8.** EMA's prediction of dimensionless horizontal location of external edge return point and comparison with experimental data.

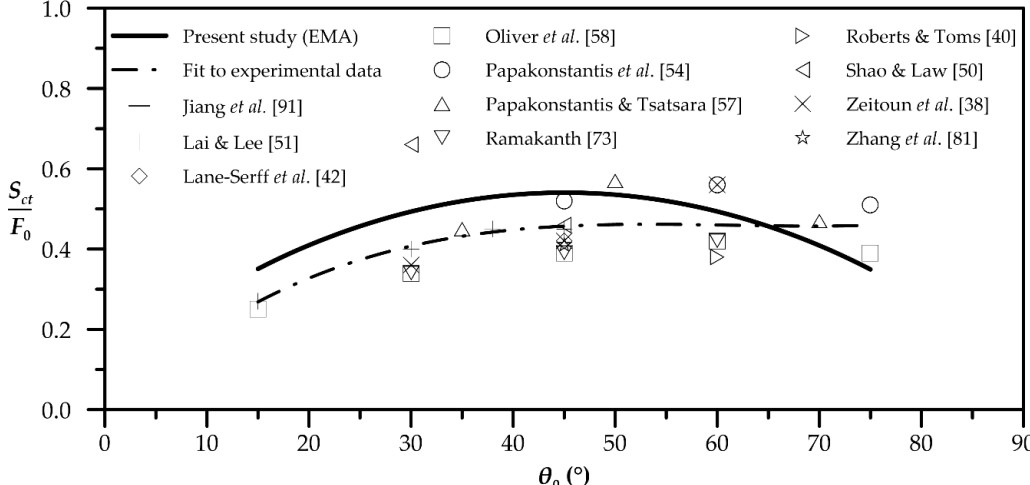

**Figure 9.** EMA's prediction of axial dilution at centerline terminal height and comparison with experimental data.

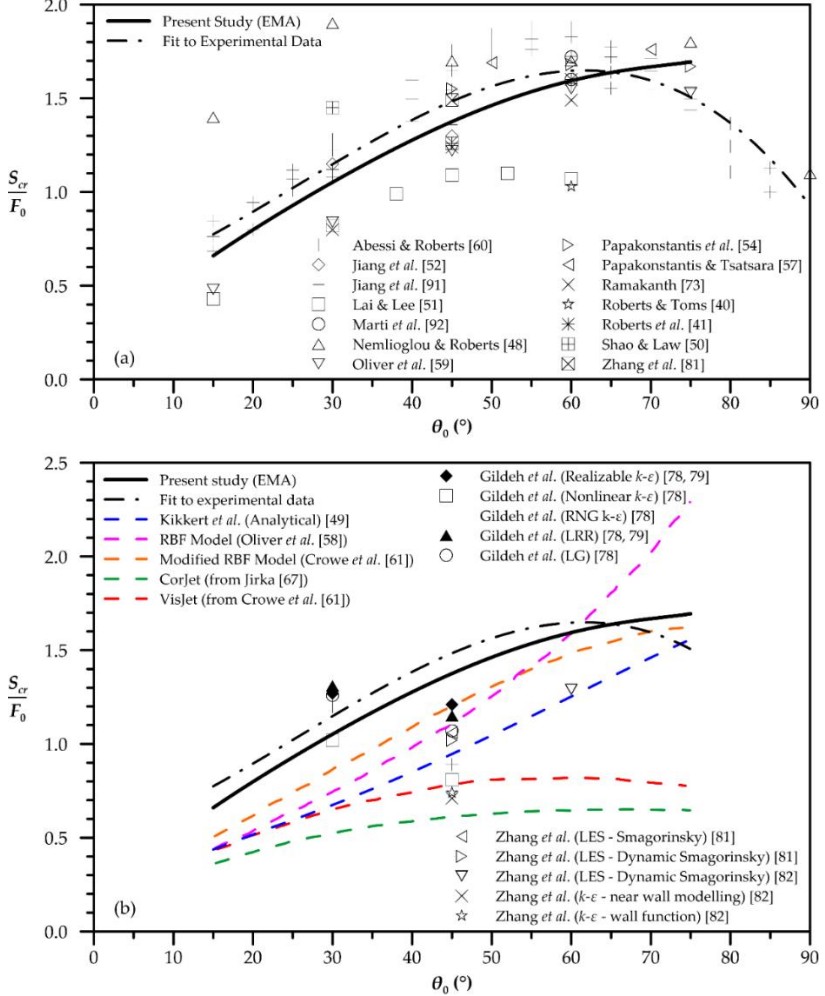

**Figure 10.** EMA's prediction of axial dilution at centerline return point and comparison with (**a**) experimental data, and (**b**) other models' predictions.

The dimensionless path length of the centerline up to terminal height $s_{ct}$ (Figure 11) almost coincides to corresponding experimental data of Crowe [69], Oliver [68] and Ramakanth [73], where the scatter of experimental data is negligible.

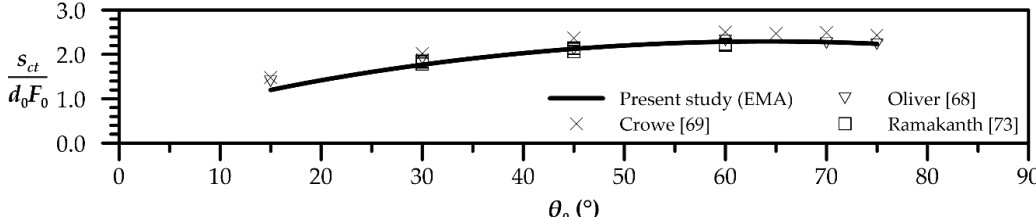

**Figure 11.** EMA's prediction of dimensionless centerline length to terminal height and comparison with experimental data.

The only existing experimental data of the path length of external edge of negatively buoyant plumes until the return point are provided by Christodoulou and Papakonstantis [95], and are compared to the corresponding predictions of EMA model in Figure 12. It is obvious that the EMA and experimental results almost coincide.

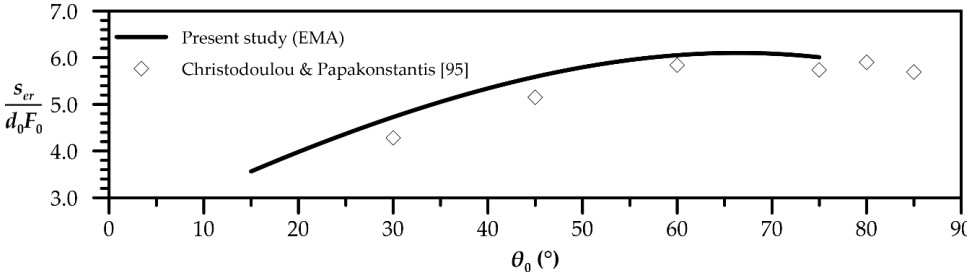

**Figure 12.** EMA's prediction of dimensionless external edge length to return point and comparison with experimental data.

## 5. Conclusions

A second order Gaussian integral model that incorporates the mechanism of escaping masses from an inclined buoyant jet (EMA) was applied to predict the mean flow and mixing properties of inclined turbulent negatively round buoyant jets in stationary and uniform ambient. The predictions were made for a range of initial inclination angle from 15° to 75° to the horizontal and initial Froude number from 4 to 100. EMA's dimensionless results were compared to experimental data and other integral or numerical predictions available in the literature. For comparison reasons, a third degree polynomial is fitted to experimental data for every parameter under study. EMA's prediction of dimensionless terminal height of external edge almost coincides to the experimental fit, while the corresponding prediction of centerline terminal height differs slightly from the polynomial fit, for initial angles $\theta_0$ from 35° to 40°. This difference is attributed to the way that centerline is defined by each researcher. EMA's predictions of horizontal distance of centerline terminal height are lower than the experimental fit polynomial, but within the range of experimental values. Regarding the horizontal location of return point of centerline and external edge, EMA's predictions follow the corresponding polynomial fits presenting a slight difference for the case of centerline variation. As an overall observation, EMA's predictions agree well to the experimental data, which assures that EMA performs better than other models. The definition of buoyant jet's centerline affects the estimated variation of axial dilution at centerline terminal height regarding the available experimental data, leading to overestimations for initial inclinations less than 45°. Additionally, EMA's prediction of the axial dilution values at centerline return point is rather conservative compared to the corresponding experimental fit, but much closer

than other models' predictions. Finally, EMA's of centerline length to terminal height and external edge length to return point coincides excellently to available experimental data. Overall, it seems that the application of EMA model gives reliable predictions without heavy computational cost or adopting controversial assumptions.

**Author Contributions:** Investigation, A.A.B. and P.C.Y.; Methodology, A.A.B. and P.C.Y.; Project administration, P.C.Y.; Software, A.A.B. and P.C.Y.; Supervision, P.C.Y.; Validation, A.A.B.; Writing–original draft, A.A.B.; Writing–review & editing, P.C.Y. All authors have read and agreed to the published version of the manuscript.

**Funding:** This research received no external funding.

**Acknowledgments:** The authors acknowledge the three reviewers' comments that improved the paper significantly.

**Conflicts of Interest:** The authors declare no conflict of interest.

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
