# Peer review of "Revisiting Mean Flow and Mixing Properties of Negatively Round Buoyant Jets Using the Escaping Mass Approach (EMA)"

_fluids, doi:10.3390/fluids5030131_

Round 1

Reviewer 1 Report

For my cooments look at the attached file

Author Response

See uploaded file.

Reviewer 2 Report

In this paper an integral model, namely EMA, is used to simulate inclined turbulent dense jet flows. This is a detailed study where the model predictions are compared with a wide range of experimental data and predictions of other models, too. The topic of the paper is interesting and within the scope of the journal. EMA model is a useful tool for predicting the environmental impact when desalination brine is discharged in the sea and for the design of outfall systems. The quality of English is very good and the paper is well written. Therefore, the paper deserves to be published.

Only few minor changes are required before publishing:

  • Keywords: “Negative buoyant jets”. I suggest the word “Negative” to be replaced by the word “Negatively”.
  • In Figures 2a, 3a, 4a, 5a, 8a the title of the horizontal axis could be added along with the values of the scale.
  • Page 5, line 217: “ … where Λ includes the value π/4 for round buoyant jets”. It could be clarified what this sentence implies.
  • Caption of Fig. 7. Modification is required as there is only one Figure (there is no Fig. 7a and Fig. 7b).

Author Response

See uploaded file.

Reviewer 3 Report

The authors present results from model calculations ("Escaping Mass Approach", EMA) that predict the mean flow development of a "negatively buoyant" (i.e. heavy) jet, which is initially oriented upwards. The predictions are compared to experimental data and to other modelling results from the literature. It is concluded that "EMA performs better than other models." In the context, this conclusion is probably meant to say "better than ALL other models." The results that are obtained from the EMA model are indeed in good agreement with experimental data, and they deserve to be published if the following points can be clarified: 1. How is the local inclination angle theta computed? This seems to be the crucial quantity, governing the curvilinear coordinate system, and therefore of highest importance for the results. 2. How are the other parameters obtained that enter the model? There are so many of them that they are difficult for me to list, especially because it is not very clear which parameters are prescribed, and which are dependent. Often the choice of parameter values is motivated by a reference to an earlier study by the authors (Ref. 74). It should be clarified for the reader to what extent the input parameters for the model rely on fitting to experimental data. Further comments: 3. The authors say in the introduction that the curvilinear system provides an "exact description of the flow field". This needs to be explained. 4. The number of 95 references is excessive for this short article.

Author Response

See uploaded file.

Round 2

Reviewer 3 Report

The authors have made a good effort to incorporate changes and reply to my remarks. I have no objections to publication.